# The Enhancing Immune Response and Anti-Inflammatory Effects of *Caulerpa lentillifera* Extract in RAW 264.7 Cells

**DOI:** 10.3390/molecules26195734

**Published:** 2021-09-22

**Authors:** Sittikorn Yoojam, Atcharaporn Ontawong, Narissara Lailerd, Kriangsak Mengamphan, Doungporn Amornlerdpison

**Affiliations:** 1Faculty of Fisheries Technology and Aquatic Resources, Maejo University, Chiang Mai 50290, Thailand; best_bby@hotmail.com; 2Division of Physiology, School of Medical Sciences, University of Phayao, Phayao 56000, Thailand; atcharaporn.on@up.ac.th; 3Department of Physiology, Faculty of Medicine, Chiang Mai University, Chiang Mai 50200, Thailand; narissara.lailerd@cmu.ac.th; 4Center of Excellence in Agricultural Innovation for Graduate Entrepreneur, Maejo University, Chiang Mai 50290, Thailand; kriang1122sak@gmail.com

**Keywords:** *Caulerpa lentillifera*, DNA damage, immune response, inflammatory cytokine, inflammatory enzyme

## Abstract

Background: *Caulerpa lentillifera* (CL) is a green seaweed, and its edible part represents added value as a functional ingredient. CL was dried and extracted for the determination of its active compounds and the evaluation of its biological activities. The major constituents of CL extract (CLE), including tannic acid, catechin, rutin, and isoquercetin, exhibited beneficial effects, such as antioxidant activity, anti-diabetic activity, immunomodulatory effects, and anti-cancer activities in in vitro and in vivo models. Whether CLE has an anti-inflammatory effect and immune response remains unclear. Methods: This study examined the effect of CLE on the inflammatory status and immune response of lipopolysaccharide (LPS)-stimulated RAW 264.7 cells and the mechanisms involved therein. RAW264.7 cells were treated with different concentrations of CLE (0.1–1000 µg/mL) with or without LPS (1 µg/mL) for 24 h. Expression and production of the inflammatory cytokines, enzymes, and mediators were evaluated. Results: CLE suppressed expression and production of the pro-inflammatory cytokines IL-6 and TNF-α. Moreover, CLE inhibited expression and secretion of the inflammatory enzyme COX-2 and the mediators PGE2 and NO. CLE also reduced DNA damage. Furthermore, CLE stimulated the immune response by modulating the cell cycle regulators p27, p53, cyclin D2, and cyclin E2. Conclusions: CLE inhibits inflammatory responses in LPS-activated macrophages by downregulating inflammatory cytokines and mediators. Furthermore, CLE has an immunomodulatory effect by modulating cell cycle regulators.

## 1. Introduction

Inflammation is a biological response to harmful stimuli including pathogens, toxic compounds, and irritants. Pro-inflammatory cytokines are produced predominantly by activated macrophages during inflammation [1]. Abundant evidence has reported that pro-inflammatory cytokines such as interleukin (IL) -6, IL-1β, and tumor necrosis factor (TNF)-α, and inflammatory enzymes including inducible nitric oxide synthase (iNOS) and cyclooxygenase-2 (COX-2), are involved in the process of inflammation [2]. Previous studies demonstrated that IL-6 has an effect on the immune system’s response due to acute inflammation [3]. Furthermore, IL-1β is important for the inflammatory response to microbial infection [4]. TNF-α is involved in the process of pathological pain [2].

The immune response is the defense mechanism of the immune system against bacteria, viruses, and substances that appear foreign and harmful. Previous studies reported that cell cycle regulators (tumor protein *p53* (p53), the cyclin-dependent kinase inhibitor *p27* (p27), cyclin D2, and cyclin E2) are involved in the immune response. The p53 gene and protein are part of the innate immune system, and play a key role in defense against infectious diseases [5]. In addition, p27 played a role in immune-mediated inflammation in a glomerulonephritis model [6]. Moreover, cell cycle proteins (cyclin D and E) are involved in the homeostasis and integrity of the immune system [7]. Thus, a potential strategy to modulate the expression and function of these inflammatory mediators is urgently needed.

*Caulerpa lentillifera* (CL) is a green seaweed, known as sea grapes or green caviar seaweed [8]. It is distributed and cultured worldwide, especially in Southeast Asia, including the Philippines, Indonesia, Vietnam, and Thailand. In the cultivation of sea grapes, more than 70% of the algae is discarded as waste. Previous studies reported that CL contains proteins, minerals, dietary fiber, vitamins, and saturated and unsaturated fatty acids [9]. In addition, CLE exhibited beneficial effects, including antioxidant activity, anti-diabetic activity, immunomodulatory effects, and anti-cancer activities [9,10,11,12]. Furthermore, sulfated polysaccharide CLGP4 from CL reduced IL-1β and TNF-α in HT29 colonic carcinoma cells [13]. However, the effect of an aqueous CL extract (CLE) on inflammatory status and its mechanisms remains unclear. Thus, we examined the anti-inflammatory effect of CLE on RAW264.7 macrophages stimulated by LPS, as well as the mechanism involved.

## 2. Results

### 2.1. Phenolic Compounds in CLE

Figure 1 shows that the LC-MS chromatogram for quantitated specification of the biocompounds in CLE reliably detected gallic acid, catechin, tannic acid, rutin, isoquercetin, and quercetin. The calculated amounts of each major constituent found in CLE, as obtained from their respective calibration curves, are shown in milligrams of the phenolic compound per gram of CLE. A high content of tannic acid was obtained (4.715 mg/g) compared with each phenolic compound standard. In addition, catechin was found at 2.629 mg/g of CLE; whereas rutin, isoquercetin, gallic acid, and quercetin were present at 0.688, 0.431, 0.390, and 0.361 mg/g of CLE, respectively.

### 2.2. Effects of CLE on Pro-Inflammatory Cytokine Secretion in RAW 264.7 Cells

The anti-inflammatory effect of CLE in RAW264.7 cells was determined using an ELISA assay. As shown in Figure 2, IL-6, IL-1β, and TNFα levels significantly increased in cells treated with 1 µg/mL LPS compared with the control. CLE at 1–1000 µg/mL significantly decreased IL-6 production compared with LPS-treated cells (Figure 2A). In addition, CLE at 1000 µg/mL reduced IL-1β compared with LPS-treated cells (Figure 2B). Likewise, CLE at 0.1–1000 µg/mL markedly reduced TNFα levels compared with LPS-treated cells (Figure 2C). Similarly, celecoxib (CX), a non-steroidal anti-inflammatory drug, also decreased pro-inflammatory cytokines. In addition, cell viability in test compounds, including LPS, 0.1–1000 µg/mL, and CX was also examined. However, CLE at high doses (100 and 1000 µg/mL) significantly reduced cell viability compared with the control cells (Figure 2D). Due to the cytotoxicity of CLE at 1000 µg/mL, CLE at 10 µg/mL was selected because it significantly decreased inflammatory cytokines without cytotoxicity.

### 2.3. Effects of CLE on the Expression of Pro-Inflammatory Cytokine Genes in RAW 264.7 Cells

To further clarify the anti-inflammatory effect of CLE, expression of pro-inflammatory cytokine genes, including IL-6, IL-1β, and TNF-α, was also investigated using real-time PCR. Pro-inflammatory cytokine gene expression was significantly reduced in CLE-treated cells compared with control cells (Figure 3). These data indicate that CLE inhibits pro-inflammatory cytokine expression at the transcriptional level in activated macrophages.

### 2.4. Effects of CLE on Inflammatory Signaling Molecules

As shown in Figure 4A, LPS significantly increased nitric oxide (NO) production compared with control cells. In addition, CLE reduced NO production levels compared with LPS-treated cells. This study then investigated the effects of CLE on inflammatory pathway molecules, including prostaglandin E2 (PGE2), a principal mediator of inflammation, and cyclooxygenase-2 (COX-2), a prostaglandin-endoperoxide synthase. The results showed that LPS elevated COX-2 gene expression and PGE2 production compared with control cells. Moreover, CLE significantly decreased COX-2 gene expression, which resulted in a reduction in PGE2 production (Figure 4B,C). Thus, this study suggests that treatment with CLE has an anti-inflammatory effect in activated macrophage cells, primarily through inhibiting COX-2 expression and PGE2 production, resulting in improved inflammatory status.

### 2.5. Effects of CLE on DNA Damage

The inhibitory effect of CLE on DNA damage was evaluated. RAW264.7 cells treated with CLE with or without LPS for 24 h were stained with Hoechst 33342, a fluorescent stain for labeling DNA. The results showed that LPS induced morphologic changes in the DNA, which presented as nuclear fragmentation, chromatin condensation, and apoptotic body formation, compared with the control cells (Figure 5B). Interestingly, these parameters were not seen in CLE-treated cells, similar to CX-treated cells (Figure 5C,D).

In addition, the effect of CLE on DNA damage was further confirmed by an ELISA assay. As shown in Figure 5E, the LPS treatment significantly increased the level of the oxidative stress-induced DNA damage marker, 8-hydroxy-2′- deoxyguanosine (8-OHdG) adduct, compared with control cells. However, CLE-treated cells markedly reduced 8-OHdG generation, similar to CX-treated cells, indicating that CLE had an inhibitory effect on the DNA damage response.

### 2.6. Effects of CLE on the Immune Response

The effect of CLE on immune response enhancement was further investigated using qPCR. As shown in Figure 6, the expression of p53, a tumor suppressor gene, was inhibited by CLE compared with the control. Similarly, p27, a cell cycle inhibitor, was also suppressed by CLE. However, the expression of cyclin D2 and cyclin E2, which are cell cycle inducer genes, was increased in CLE-treated cells. These data suggest that CLE stimulated the immune response by modulating cell cycle regulators.

## 3. Discussion

Inflammation is a response of the immune system that can be stimulated by several factors, including pathogens, damaged cells, and toxic compounds. Dysregulation of the inflammatory response is associated with the development of sepsis. Inflammation is a major cause of increased morbidity and mortality in intensive care units [14]. Inflammatory cells release pro-inflammatory cytokines, such as IL-1, IL-6, and IL-12, and inflammatory enzymes, including iNOS and COX-2, driving activation of immune cells [14]. Thus, improved inflammatory status through reduced inflammatory cytokines and enzymes could be potential targets for the prevention of sepsis. This study aimed to evaluate the anti-inflammatory and immune-promoting effects of a *Caulerpa lentillifera* extract (CLE) and identify their mechanisms in RAW 264.7 cells. The current study clearly showed that CLE exhibited anti-inflammatory effects in LPS-stimulated RAW264.7 cells by decreasing IL6 and TNF-α production and expression of the related genes. Consistently, previous studies found that the major constituent in CLE, tannic acid, suppressed IL-6 and IL-1β production and also reduced TNF-α protein expression [15]. Accordingly, the catechin found in CLE also showed suppression of the inflammatory response by inhibiting the expression of pro-inflammatory cytokine genes, including IL-1α, IL-1β, IL-6, and IL-12p35, and enhancing the expression of anti-inflammatory cytokine genes, including IL-4 and IL-10 in 3T3-L1 adipocytes [16]. Thus, the anti-inflammatory effects of CLE seen in this study might be due to tannic acid and catechin.

During inflammation, iNOS enzymes produce high levels of NO, which has been determined to be a cytotoxic molecule. In addition, prostaglandin G2 is converted into prostaglandin H2 by COX enzymes and then activated prostaglandin synthases to generate PGE2, a principal mediator of inflammation. Moreover, pro-inflammatory cytokines are important mediators in the transcription of iNOS and COX-2 genes [17,18,19]. The results of the present study demonstrated that CLE decreased NO compared with the control. CLE also reduced PGE2 levels and COX-2 enzyme, similar to the action of NSAIDs. Previous studies have reported that condensed tannin modulated the expression of inflammatory mediator enzymes, including iNOS and COX-2, in a mouse model [20]. It was reported that the hydrophobic “core” and the hydrophilic “shell” of tannic acid are the features responsible for its antioxidant and anti-inflammatory action [21]. Furthermore, catechin also inhibited the expression of inflammatory enzyme genes, including iNOS and COX-2, in 3T3-L1 adipocytes [16], which suggests that CLE has potential as an anti-inflammatory agent.

Inflammation produces reactive oxygen and nitrogen species (RONS), which can stimulate tissue repair and regeneration; however, the chemicals from this process can also induce DNA damage. This study showed that LPS induced DNA damage by nuclear fragmentation, chromatin condensation, and apoptotic body formation. Nevertheless, CLE reversed DNA damage, similar to NSAIDs. Consistently, complex polyphenols and tannins from wine decreased DNA oxidative damage in mucosal cells in rat colon [22]. Previous studies also demonstrated that hydroxyl radicals induced supercoiled plasmid DNA to break into a linear/open circular form [23]. However, condensed tannins from *V. angularis* seeds inhibited supercoiled plasmid DNA damage [24]. Similarly, EGCG, the most active catechin, reduced the level of prompt DNA strand breaks [25].

Cell proliferation leads to an exponential increase in cell numbers, resulting in an increased tissue growth rate. Some of the important cell proliferation regulators are the D cyclins (cyclins D1, D2, and D3), which regulate cell cycle progression. Furthermore, during the late G1 stage, E cyclins (cyclins E1 and E2) become upregulated and activate cyclin-dependent kinase (CDK)-2, resulting in phosphorylation of various cell cycle-related proteins [7]. In addition, p53 and p27 are direct inhibitors of cell cycle progression [26]. This study first demonstrated that an increase in proliferation occurred in RAW 264.7 cells treated with CLE. The expression of cell cycle regulators, including cyclin D2 and cyclin E2, was increased by treatment with CLE, while the expression of p53 and p27 was inhibited. Moreover, other natural products could promote the immune response via regulating cell cycle regulators. *Anemarrhena asphodeloides* extract activated the immune response by extending the cell cycle S-phase, inhibiting p27 and elevating cyclin D2 and cyclin E2 gene expression in RAW264.7 cells [27].

## 4. Materials and Methods

### 4.1. Chemicals

Dulbecco’s modified Eagle’s medium (DMEM) and fetal bovine serum (FBS) were purchased from Gibco (Carlsbad, CA, USA). β-Nicotinamide adenine dinucleotide phosphate; (3-4, 5-dimethylthiazol-2-yl-2),5-diphenyltetra-zolium bromide (MTT); lipopolysaccharides were purchased from Merck (Darmstadt, Germany). ELISA kits were obtained from BioLegend (San Diego, CA, USA). All other chemicals with high purity were obtained from commercial sources.

### 4.2. Preparation and Chemical Characterization of CLE

Low-grade CL was obtained from Phetchaburi Coastal Fisheries Research and Development Center, Phetchaburi Province, Thailand. A voucher specimen (number AI (2160 has been deposited at the herbarium of the Center of Excellence in Agricultural Innovation for Graduate Entrepreneur, Maejo University, Chiang Mai, Thailand. The fresh seaweed was rinsed and dried in an oven. Dried CL was soaked in distilled water and boiled at 100–95 °C for 2 h. The aqueous solution was filtered through filter paper. It was then concentrated using a rotary evaporator and lyophilized by freeze-drying. The CLE was stored at 4 °C prior to determination of the active compounds and evaluation of its biological activities. In addition, CLE was evaluated for its phenolic compound profile using liquid chromatography–mass spectrometry (LC-MS) (Agilent 1100 series, LC/MSD SL, Waldbronn, Germany).

### 4.3. Cell Culture

Murine RAW264.7 cells were purchased from the American Type Culture Collection (ATCC) (Manassas, VA, USA). Cells in the 2nd–22nd passages were grown in Dulbecco’s modified Eagle’s Medium (DMEM) (Life Technologies, New York, NY, USA) containing 3.7 g/L NaHCO_3_ supplemented with 10% fetal bovine serum (FBS) and 1% penicillin–streptomycin in a humidified atmosphere containing 5% CO_2_. RAW264.7 cells were seeded in 12- and 96-well plates at a density of 2.0 × 105 cells/mL and grown for 3 days for further experiments.

### 4.4. Determination of Cell Viability

The viability of RAW264.7 cells was determined using a 3-(4,5-dimethylthiazol-2-yl)-2,5-diphenyltetra-zolium bromide (MTT) assay. Cells were seeded at 2.0 × 105 cells/mL in 96-well plates. The cells were incubated in a culture medium with or without various concentrations of CLE (0.1–1000 µg/mL) or celecoxib (CX), a non-steroidal anti-inflammatory drug (NSAID), at 3.8 µg/mL for 24 h at 37 °C. After exposure, the cells were washed with PBS, and free DMEM containing a MTT reagent was added and incubated for the next 4 h at 37 °C. Finally, the MTT solution was aspirated and DMSO was added to each well and incubated for another 30 min at 37 °C. The viability of RAW264.7 cells was measured at a wavelength of 570 nm using a SynergyTM HT microplate reader (Biotek, Winooski, VT, USA). The sample detected at a wavelength of 680 nm was also used as a reference.

### 4.5. ELISA Assay

The cells were seeded in 12-well plates at 2.0 × 105 cells/mL and cultured for 3 days at 37 °C. On the day of the experiment, 1 µg/mL of LPS or CX at 3.8 µg/mL was added to the culture medium with or without various concentrations of CLE (0.1–1000 µg/mL) for 24 h at 37 °C. The treated cells were then taken for the next assay and centrifuged at 2000× *g* for 10 min at 4 °C. The supernatant was collected for determining the IL-6, IL-1β, TNF-α, NO, and PGE2 concentrations using commercial kits, according to the manufacturer’s instructions (BioLegend, San Diego, CA, USA). The levels of inflammatory cytokines were measured at a wavelength of 540 nm using a SynergyTM HT microplate reader (Biotek, Winooski, VT, USA).

### 4.6. Nitric Oxide Assay

The cells were plated in a 12-well plate at 2.0 × 105 cells/mL and cultured for 3 days at 37 °C. On the day of the experiment, culture media with or without various concentrations of CLE (0.1–1000 µg/mL) or CX at 3.8 µg/mL, with or without 1 µg/mL of LPS, were added for 24 h at 37 °C. The treated cells were then taken for the next assay and centrifuged at 10,000× *g* for 20 min at 4 °C. The supernatant was subsequently collected for measurement of the nitric oxide concentration. The nitric oxide level was measured at a wavelength of 540 nm using a SynergyTM HT microplate reader (Biotek, Winooski, VT, USA).

### 4.7. DNA Damage Assay

To further identify the effect of CLE on DNA damage, the effect of CLE on 8-hydroxy-2′- deoxyguanosine (8-OHdG), a DNA damage marker, was investigated using an ELISA assay. RAW264.7 cells were plated in 12-well plates (2 × 105 cells/mL) and incubated for 24 h at 37°C in a humidified atmosphere with 5% CO_2_. The culture medium was removed, then the cells were supplemented with various concentrations of CLE (0.1–1000 µg/mL) or CX at 3.8 µg/mL, with or without 1 µg/mL of LPS, for 24 h at 37 °C. The treated cells were then taken for the next assay and centrifuged at 10,000× *g* for 20 min at 4 °C. The supernatant was subsequently collected for measurement of the 8-OHdG concentration using commercial kits, according to the manufacturer’s instructions (Thermo Fisher Scientific, Waltham, MA, USA).

### 4.8. Hoechst 33,342 Staining

To confirm the effect of CLE on DNA damage, the cells were seeded in 8-well cell culture slides for 3 days. RAW264.7 cells were supplemented with various concentrations of CLE (0.1–1000 µg/mL) or CX at 3.8 µg/mL, with or without 1 µg/mL of LPS, for 24 h at 37 °C in a humidified atmosphere with 5% CO_2_. The treated cells were fixed with 4% paraformaldehyde for 10 min and then treated with Hoechst 33,342 staining at 5 µg/mL for 10 min. The cells were then washed twice with PBS and observed under a Nikon Eclipse Ni-U fluorescent microscope (Nikon).

### 4.9. Quantitative Real-Time PCR Analysis

Total RNA was extracted and purified from RAW264.7 cells using TRIzol reagent (Thermo Fisher Scientific, Waltham, MA, USA), according to the manufacturer’s instructions. First-strand cDNA was obtained using a SensiFAST cDNA synthesis kit (Bioline, London, UK), and qPCR was performed using SYBR Real-Time PCR Master Mix (Bioline, London, UK) on a CFX Touch real-time PCR system (Bio-Rad, Hercules, CA, USA). Forward and reverse primers were purchased from Macrogen (Seoul, Korea) and used at a final concentration of 0.4 µM. The specific primer sets for mouse TNF-α, IL-1β, IL-6, COX-2, p53, p27, cyclin D2, cyclin E2, and GAPDH that were used for the qRT-PCR are presented in Table 1. Gene expression was normalized to GAPDH levels and is reported as relative fold changes (RFC) [28,29,30,31,32,33]. The qPCR amplifications were performed in duplicate for each cDNA.

### 4.10. Statistical Analysis

Data are expressed as the mean ± S.E.M. Statistical differences were assessed using one-way ANOVA followed by Dunnett’s test. Statistical analyses were conducted using the Statistical Package for the Social Sciences version 23 (IBM Corp., New York, NY, USA). Differences were considered significant at *p* < 0.05.

## 5. Conclusions

CLE contains tannin and catechin, and it improves inflammatory status by inhibiting the gene expression and production of pro-inflammatory cytokines and their mediators, and also by suppressing apoptotic body formation and DNA damage. CLE also enhances the immune response by modulating cell cycle regulators.

## Figures and Tables

**Figure 1 molecules-26-05734-f001:**
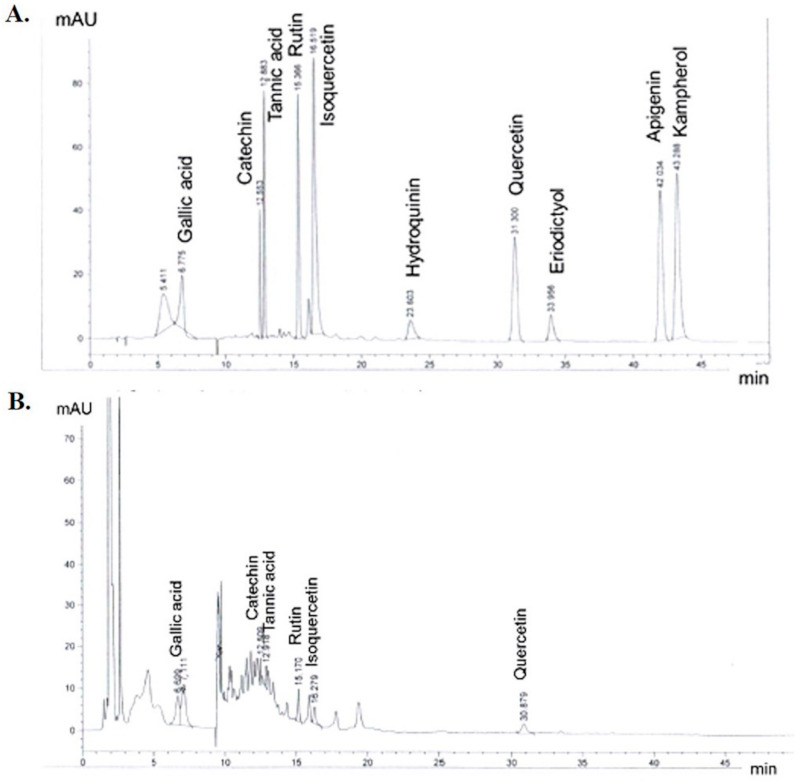
A representative LC-MS chromatogram of (**A**) the standards of the phenolic compounds, and (**B**) gallic acid, catechin, tannic acid, rutin, isoquercetin, and quercetin contained in *Caulerpa lentillifera extract* (CLE).

**Figure 2 molecules-26-05734-f002:**
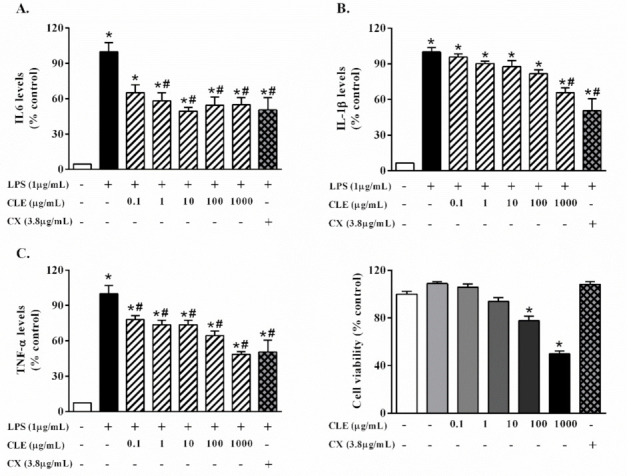
The effects of CLE on inflammatory cytokine production in LPS-stimulated RAW 264.7 cells. RAW 264.7 cells were treated with LPS in the absence or presence of different concentrations of CLE for 24 h. The production of (**A**) IL-6, (**B**) IL-1β, and (**C**) TNF-α was measured by ELISA. (**D**) Viability of RAW 264.7 cells after exposure to different concentrations of CLE with or without LPS for 24 h. Values shown are mean ± S.E.M. (*n* = 5); * *p* < 0.05 vs. control and # *p* < 0.05 vs. LPS.

**Figure 3 molecules-26-05734-f003:**
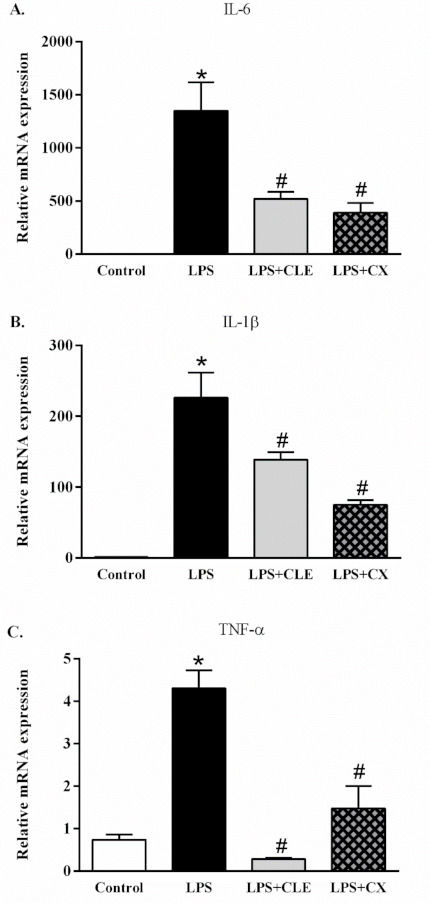
The effects of CLE on the expression of inflammatory cytokine genes in LPS-stimulated RAW 264.7 cells. RAW 264.7 cells were treated with CLE at 10 µg/mL in the presence or absence of LPS for 24 h. The expression of (**A**) IL-6, (**B**) IL-1β, and (**C**) TNF-α was measured by qPCR. Values shown are mean ± S.E.M. (*n* = 5); * *p* < 0.05 vs. control and # *p* < 0.05 vs. LPS.

**Figure 4 molecules-26-05734-f004:**
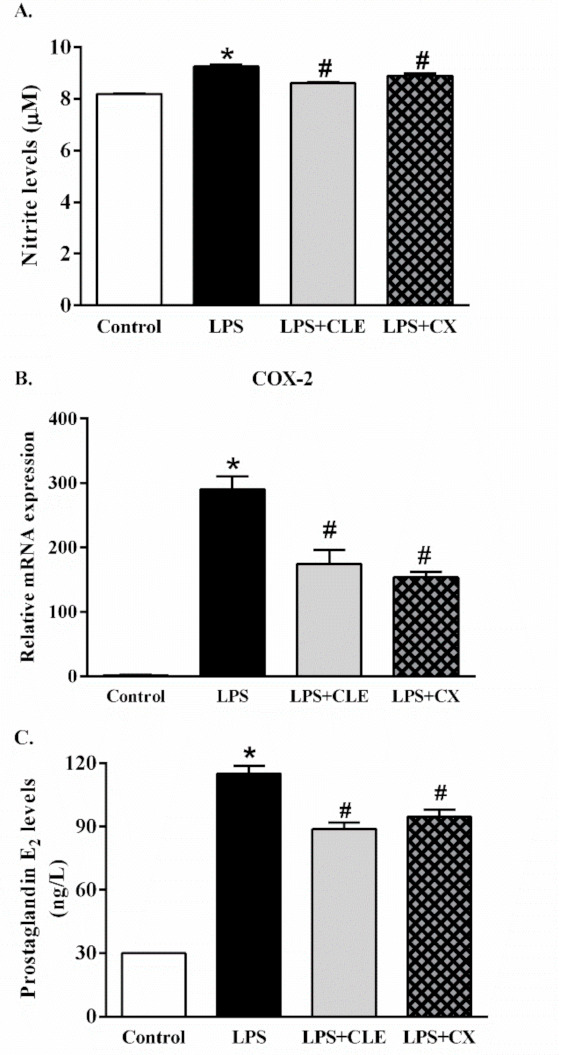
The effects of CLE on NO production, COX2 gene expression, and PGE2 production in LPS-stimulated RAW 264.7 cells. RAW 264.7 cells were supplemented with CLE or CX in the presence or absence of LPS for 24 h. The production of NO (**A**), the expression of cyclooxygenase-2 (COX-2) (**B**), and prostaglandin E2 (PGE2) levels (**C**) were evaluated. Values shown are mean ± SEM (*n* = 5); * *p* < 0.05 vs. control and # *p* < 0.05 vs. LPS.

**Figure 5 molecules-26-05734-f005:**
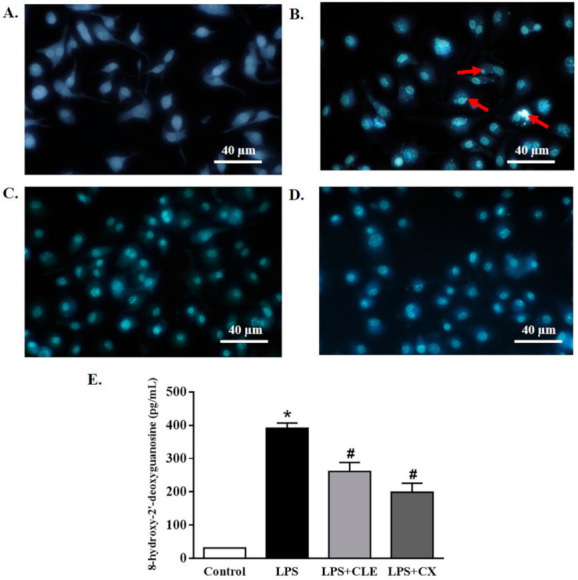
The effects of CLE on DNA damage in LPS-stimulated RAW 264.7 cells. Cells were plated in 12-well plates and treated with CLE or CX in the presence or absence of LPS. Cells were treated 24 h later with Hoechst 33,342 staining at 5 µg/mL for 10 min and then observed under an inverted fluorescence microscope (original magnification, ×40). The change in the nucleus of apoptotic cells is shown by the arrows (**A**–**D**). The amount of 8-OHdG in the DNA was determined using an 8-OHdG-EIA kit (**E**). Values shown are mean ± SEM (*n* = 5); * *p* < 0.05 vs. control and # *p* < 0.05 vs. LPS.

**Figure 6 molecules-26-05734-f006:**
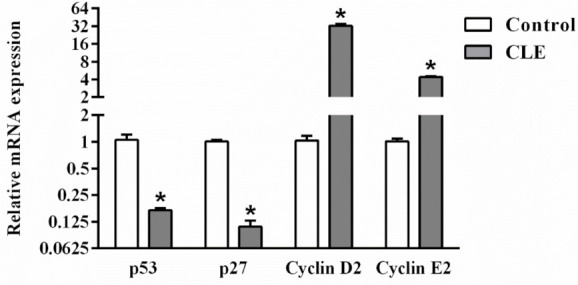
The effects of CLE on cell regulator molecules in LPS-stimulated RAW 264.7 cells. The expression of cell cycle regulator genes was determined by real-time PCR after cells were treated with CLE for 24 h. Values shown are mean ± SEM (*n* = 5); * *p* < 0.05 vs. control.

**Table 1 molecules-26-05734-t001:** Primer sequences and expected amplicon sizes for gene amplification.

cDNA	Genbank Acc. No.	Forward Primer	Reverse Primer	Amplicon Size (bp)
TNF-α	NM013693.3	5′-ACCTGGCCTCTCTACCTTGT-3′	5′-CCCGTAGGGCGATTACAGTC-3′	161
IL-1β	NM008361.4	5′-GCCACCTTTTGACAGTGATGAG-3′	5′-AGTGATACTGCCTGCCTGAAG-3′	165
IL-6	NM031168.2	5′-CAACGATGATGCACTTGCAGA-3′	5′-TCTCTCTGAAGGACTCTGGCT-3′	201
COX-2	NM011198.4	5′-CCACTTCAAGGGAGTCTGGA-3′	5′-AGTCATCTGCTACGGGAGGA-3′	197
CyclinD2	NM009829.3	5′-ACCTCCCGCAGTGTTCCTATT-3′	5′-CACAGACCTCTAGCATCCAGG-3′	93
CyclinE2	NM001037134.2	5′-TCTGTGCATTCTAGCATCGACTC-3′	5′-AAGGCACCATCGTCTACACATTC-3′	149
p27	NM009875.4	5′-GCGGTGCCTTTAATTGGGTCT-3′	5′-GGCTTCTTGGGCGTCTGCT-3′	230
p53	NM011640.3	5′-ACCGCCGACCTATCCTTACC-3′	5′-TCTTCTGTACGGCGGTCTCTC-3′	118
GAPDH	NM001289726.1	5′-TGTGTCCGTCGTGGATCTGA-3′	5′-TTGCTGTTGAAGTCGCAGGAG-3′	150

## Data Availability

The datasets analyzed during the current study are available from the corresponding author on reasonable request.

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
