# Peer review of "The Enhancing Immune Response and Anti-Inflammatory Effects of Caulerpa lentillifera Extract in RAW 264.7 Cells"

_molecules, 2021, doi:10.3390/molecules26195734_

Round 1

Reviewer 1 Report

I have carefully read the manuscript, which I found very interesting and significant for the scientific community. In my opinion, this study was well organized and conducted. Manuscript is written in very good language. Therefore, I have no requests for the authors and recommend the acceptance in the present form.

Author Response

The manuscript has undergone English language editing by MDPI. The text has been checked for correct use of grammar and common technical terms, and edited to a level suitable for reporting research in a scholarly journal. 

Reviewer 2 Report

This manuscript is about the anti-inflammatory of caulerpa lentillifera. Unfortunately, similar content has been published in the international journal of biological macromolecules 2020. The approach of this manuscript does not seem to be improved comparing with the reported paper. The manuscript would be improved if the authors include the mode of action of caulerpa lentillifera in the manuscript.

Author Response

We greatly appreciate the reviewer’s comment which have indeed helped to improve the quality of this manuscript. 

The previous study found that sulfated polysaccharide of Caulerpa lentillifera possessed anti-inflammatory activity and immunostimulatory activity (Sun et al, 2020; Zhang et al, 2020). Moreover, the aqueous extract of  Caulerpa lentillifera was evaluated for signaling molecules and DNA damage on inflammatory status and immune response in this study.

Sun Y, Liu Z, Song S, Zhu B, Zhao L, Jiang J, Liu N, Wang J, Chen X. Anti-inflammatory activity and structural identification of a sulfated polysaccharide CLGP4 from Caulerpa lentillifera. Int J Biol Macromol. 2020 Mar 1;146:931-938. doi: 10.1016/j.ijbiomac.2019.09.216. Epub 2019 Nov 12. PMID: 31730965

Zhang M, Zhao M, Qing Y, Luo Y, Xia G, Li Y. Study on immunostimulatory activity and extraction process optimization of polysaccharides from Caulerpa lentillifera. Int J Biol Macromol. 2020 Jan 15;143:677-684. doi: 10.1016/j.ijbiomac.2019.10.042. Epub 2019 Nov 12. PMID: 31730975.

We agree with the reviewer that Introduction” requires improvement. Therefore, we have rewritten as suggested in “Introduction” section in line 45-54  and added these 3 additional references in “References” section.

Reviewer 3 Report

The authors of the manuscript “The enhancing immune response and anti-inflammatory effects of Caulerpa lentillifera extract in RAW 264.7 cells” studied the effect of Caulerpa lentillifera extract (CLE) in RAW 264.7 cells.

Figure 1.

The authors start the manuscript by presenting the composition of phenolic compounds in CLE. However, they do not offer any follow up of the figure, such as pinpointing the exact molecule(s) that are responsible for the anti-inflammatory effect, or anything in line with that. What is the purpose of Figure 1?

Figure 2.

The authors show the effect of CLE on LPS stimulation of cells. Data representation is confusing – were cells pre-treated with CLE or (as figure demonstrates) separately treated with either LPS or increasing concentrations of CLE?

Figure 3.

The authors state that “RAW 264.7 cells were pre-treated with different concentrations of CLE in the presence or absence of LPS for 24 h”. I believe that this is wrong (probably copy-posted from Figure 2) and that the authors only use one CLE concentration with or without LPS. Correct?

Figure 4.

The authors claim that “LPS significantly increased nitric oxide (NO) production compared with control cells. In addition, CLE markedly reduced NO production level compared with LPS-treated cells.”

However, the difference in NO production in LPS- and LPS/CLE-treated cells is very mild. The authors should adjust their statement.

What about mRNA levels of iNOS in the same experimental setup?

How do the authors explain the effect/suggest mechanism of CLE on transcription of cytokines and COX-2?

Figure 5.

What does Figure 5A represent? The authors should quantify the number of apoptotic cells and present additional apoptotic marker (such as Casp3 cleavage by Western blot).

Also, the authors should stain the cells with gamma-H2AX in order to measure DNA damage/repair.

The authors claim that „CLE also improved DNA damage (Lane 26).

What does that even mean? That there is more DNA damage upon cell treatment with CLE? The statement is very confusing.

What does this sentence mean: “This study showed that LPS induced DNA damage by nuclear fragmentation, chromatin condensation and apoptotic body formation.” How the reviewer understands the sentence, LPS is inducing DNA damage through nuclear fragmentation, chromatin condensation and apoptotic body formation, which is confusing.

Lane 237: “CLE had an inhibitory effect on DNA damage response.”

That sentence implies that CLE prevents DNA repair upon DNA damage. The sentence is misleading. Furthermore, the authors do not offer any data about the mechanism of CLE, i. e. if CLE prevents DNA damage or somehow stimulates DNA damage repair. The authors need to be careful what they state based on their results.

Figure 6.

The authors claim to study the immune response by measuring mRNA levels of p53, p27, cyclin D2 and cyclin E2. How exactly is that “immune response”? What about protein levels of p53 upon CLE treatment?

What about p53 effect on apoptosis and DNA repair?

Even though the authors show data suggesting the potential anti-inflammatory effect of CLE, they do not provide any mechanism of the described effect.

The effect is also measured in only one cell line making it unclear if it is cell line-specific or more general. Furthermore, higher CLE doses are highly cytotoxic and therefore the claim that “This magnificent effect of CLE makes it a new alternative approach as a natural anti-inflammation supplement.” is a clear overstatement and should be removed.

There are many spelling and grammatical errors in the text, such as:

Lane 18: ant-diabetic

Lane 91: Cells were seed…

Lane 144: …concentration of 0.4 (weird symbol) M.

Lane 173: markly

Superscript is missing throughout the text (examples lanes 91, 101, 111).

Moreover, sentences are hard to understand, making the following of the manuscript very difficult.

Examples (unclear what sentence means):

Lane 25: Moreover, CLE inhibited inflammatory enzymes and mediator including COX-2, PGE2 and NO, expression and secretion, respectively.

Lane 47: TNF-α is play a key role in pain models [9].

Lane 119: To further identify the protective effect of CLE on LPS induced DNA damage.

Lane 130: To confirm the effect of CLE on LPS induced DNA damage.

Lane 169: As shown in Figure 2, LPS at 1 μg/mL significantly increased pro-inflammatory cytokine; IL-6, IL-1β, and TNFα compared control cells.

Because of the superficial editing, some words are repeated in the sentences:

Lane 122: The culture medium was removed, then the cells were supplemented with various concentrations various concentrations of CLE (0.1-1000 μg/mL) or CX at 3.8 μg/mL with or without 1 μg/mL of LPS for 24 h at 37 °C.

In conclusion, the whole text has to be significantly edited by native English speaker, both for clarity and grammar/spelling.

Author Response

Response to Referee’s Comments

We greatly appreciate the reviewer’s comment which have indeed helped to improve the quality of this manuscript. The detail is present as suggested in a newly revised manuscript and the changes have made as highlighted in “red font”. We hope our revision has improved the paper to a satisfactory level.

Major comments:

  1. Figure 1

As suggested, we re-written the figure legend as follow;

Figure 1 A representative LC-MS chromatogram of (A) the standards of the phenolic compounds, and (B) gallic acid, catechin, tannic acid, rutin, isoquercetin, and quercetin contained in Caulerpa lentillifera extract (CLE).

Result: Figure 1 shows that the LC-MS chromatogram for quantitated specification of the biocompounds in CLE reliably detected gallic acid, catechin, tannic acid, rutin, isoquercetin, and quercetin. (line 70-71)

  1. Figure 2

As suggested, we have re-written the figure legend as follow;

Figure 2. The effects of CLE on inflammatory cytokine production in LPS-stimulated RAW 264.7 cells. RAW 264.7 cells were treated with different concentrations of CLE in the presence or absence of LPS for 24 h. (line 98-99)

  1. Figure 3

As suggested, we have re-written the figure legend as follow;

Figure 3. The effects of CLE on the expression of inflammatory cytokine genes in LPS-stimulated RAW 264.7 cells. RAW 264.7 cells were treated with CLE at 10 µg/mL in the presence or absence of LPS for 24 h. (line 114-115)

  1. Figure 4

We agree with the reviewer that result of Figure 4 should be softened the sentense. Thus, we have cut the word “markedly” as suggested. Inducible nitric oxide synthase (iNOS) is key enzymes generating nitric oxide (NO) in macrophage. In addition, this study we have assessed effect of CLE on NO levels that can be referred to iNOS activity, thus, we did not investigated iNOS mRNA expression.

Previous study showed that COX-2 is expressed by inflammatory cells and can be induced by NO. Therefore, this study also investigated effect of CLE on COX-2 mRNA expression. In addition, we have re-analysed the data and the result showed that CX treated group also reduced NO level compared to LPS group, thus, we have added sharp symbol (#) on CX bar graph in Figure 4. (page 6)

  1. Figure 5
  • What does Figure 5 represent?

Figure 5 is represent the effect of CLE on LPS induced DNA damage. In this study, we used Hoechst 33342 to stained the morphologic changes of DNA similar to previously described [1,2]. In addition, we agree with the reviewer that we did not showed cell apoptosis markers. Thus, we have rewritten as suggested in “Result” section in line 144-146.

References

[1] Xu XF, Zhang TL, Jin S, et al. Ardipusilloside I induces apoptosis by regulating Bcl-2 family proteins in human mucoepidermoid carcinoma Mc3 cells. BMC Complement Altern Med. 2013; 13:322.

[2] Huang F, Liu Q, Xie S, Xu J, Huang B, Wu Y, Xia D. Cypermethrin Induces Macrophages Death through Cell Cycle Arrest and Oxidative Stress-Mediated JNK/ERK Signaling Regulated Apoptosis. Int J Mol Sci. 2016;17(6):885. 

  • CLE improved DNA damage.

We agree with the reviewer that this statement is confusing, therefore, we have rewritten as suggested in “Abstract” section in line 26-27.

  • What does this sentence mean. “This study showed that LPS induced DNA damage by nuclear fragmentation, chromatin condensation and apoptotic body formation.

Figure 5B showed  LPS induced morphologic changes of DNA by increasing DNA fragmentation and  nuclear condensation (red arrow) similar to previously described [1,2]. Thus, we have rewritten as suggested in “Results” section in line 144-146.

References

[1] Xu XF, Zhang TL, Jin S, et al. Ardipusilloside I induces apoptosis by regulating Bcl-2 family proteins in human mucoepidermoid carcinoma Mc3 cells. BMC Complement Altern Med. 2013; 13:322.

[2] Huang F, Liu Q, Xie S, Xu J, Huang B, Wu Y, Xia D. Cypermethrin Induces Macrophages Death through Cell Cycle Arrest and Oxidative Stress-Mediated JNK/ERK Signaling Regulated Apoptosis. Int J Mol Sci. 2016;17(6):885. 

  • CLE had an inhibitory effect on DNA damage response.

 We agree with the reviewer that this sentence should be re-writed. Thus, we have rewritten as suggested in “Results” section in line 144-146.

  1. Figure 6
  • Immune response

The immune response is the defense mechanism of the immune system against bacteria, viruses, and substances that appear foreign and harmful. Previous studies reported that cell cycle regulators (tumor protein p53 (p53), the cyclin-dependent kinase inhibitor p27 (p27), cyclin D2, and cyclin E2) are involved in the immune response. The p53 gene and protein are part of the innate immune system, and play a key role in defense against infectious diseases [5]. In addition, p27 played a role in immune-mediated inflammation in a glomerulonephritis model [6]. Moreover, cell cycle proteins (cyclin D and E) are involved in the homeostasis and integrity of the immune system [7]. Thus, a potential strategy to modulate the expression and function of these inflammatory mediators is urgently needed. (line 45-54)

Several studies indicate that p53 activation induced cell-cycle arrest and apoptosis. Moreover, this study showed that CLE reduced p53 gene expression suggesting that CLE might be inhibited cell apoptosis.

  • CLE makes it a new alternative approach as a natural anti-inflammation supplement

We agree with the reviewer that this statement should be removed. Thus, we have removed as suggested.

References

[5] Levine AJ. P53 and The Immune Response: 40 Years of Exploration-A Plan for the Future. Int J Mol Sci. 2020;21(2):541.

[6] Ophascharoensuk V, Fero ML, Hughes J, Roberts JM, Shankland SJ. The cyclin-dependent kinase inhibitor p27Kip1 safeguards against inflammatory injury. Nat Med. 1998;4(5):575-80.

[7] Laphanuwat P, Jirawatnotai S. Immunomodulatory Roles of Cell Cycle Regulators. Front Cell Dev Biol. 2019;7:23.

[8] Chen J. The Cell-Cycle Arrest and Apoptotic Functions of p53 in Tumor Initiation and Progression. Cold Spring Harb Perspect Med. 2016;6(3):a026104.

  1. Typo errors

We have checked and edited typo errors throughout the manuscript as suggested.

  1. Superscript

We have checked and edited superscript throughout the manuscript as suggested.

  1. Sentences are hard to understand

We agree with the reviewer and we have rewritten some sentences as follow;

  • Line 25-27: From “Moreover, CLE inhibited inflammatory enzymes and mediator including COX-2, PGE2 and NO, expression and secretion, respectively.” to “Moreover, CLE inhibited expression and secretion of the inflammatory enzyme COX-2 and the mediators PGE2 and NO.”
  • Line 43-44: From “TNF-α is play a key role in pain models [9].” to “TNF-α is involved in the process of pathological pain [2].”
  • Line 283: From “To further identify the protective effect of CLE on LPS induced DNA damage.” to “To further identify the effect of CLE on DNA damage.”
  • Line 294: From “To confirm the effect of CLE on LPS induced DNA damage.” to “To confirm the effect of CLE on DNA damage.”
  • Line 81-82: From “As shown in Figure 2, LPS at 1µg/mL significantly increased pro-inflammatory cytokine; IL-6, IL-1β, and TNFα compared control cells.” to “As shown in Figure 2, IL-6, IL-1β, and TNFα levels significantly increased in 1µg/mL LPS treated cells compared to that of control.”
  •  

*The manuscript has undergone English language editing by MDPI. The text has been checked for correct use of grammar and common technical terms, and edited to a level suitable for reporting research in a scholarly journal.

  1. Repeat sentences
  • Line 286-288: From “The culture medium was removed, then the cells were supplemented with various concentrations various concentrations of CLE (0.1-1000 µg/mL) or CX at 3.8 µg/mL with or without 1 µg/mL of LPS for 24 h at 37 °C.” to “The culture medium was removed, then the cells were supplemented with various concentrations of CLE (0.1-1000 µg/mL) or CX at 3.8 µg/mL with or without 1 µg/mL of LPS for 24 h at 37 °C.”

Reviewer 4 Report

Manuscript ID: molecules-1340383
Type of manuscript: Article
Title: “The enhancing immune response and anti-inflammatory effects of Caulerpa lentillifera extract in RAW 264.7 cells”

Overview and general recommendation: the manuscript deals with elucidation of anti-inflammatory and cytotoxic mechanism of action of Caulerpa lentillifera extracts, an edible seaweed. This is an interesting study, however some items should be clarified.

General comments:

  • All Latin names should be written using Italic front.
  • English language should be checked and corrected.

Specific comments:

  • Abstract:

line 23: “The inflammatory cytokines, enzymes, and mediators were evaluated.” – the presence or production or both?

line 24: the expression of which genes were suppressed?

line 27: what is connection between immune response and proteins involved in regulation of cell cycle progression?

The conclusion in the Abstract is not justified by results summarized in this section.

Abstract requires substantial improvements.

  • Introduction:

lines 35-48: why this part concerns the topic of the manuscript, especially the part about sepsa?

This section should be re-written to clarify why anti-inflammatory mechanism of action drew attention. Moreover, it is not clear if why proteins involved in regulation of cell cycle progression, that are mentioned in Abstract, were investigated. The aim of the study should be clarified.

  • Materials and Methods:

This section should be formatted according to this journal requirements, that is enumerated properly.

line 99: ELISA is an acronym and should be written using Capital letters in the whole text;

line 144: pleas correct unknown symbol in expression “0.4 ? M.”

line 149: what S.E.M. stands for?

  • Results:

This section should be formatted according to this journal requirements, that is enumerated properly.

line 159: why the concentration of determined compounds is expressed in mg/kg? more frequently µg or mg/g dry weight are used;

and this sentence is a little bit awkward, please re-write eg. In addition, catechin content amount to … etc.

line 169: what was significantly increased? content? secretion rate?

line 170: please remove “;” moreover it is not clear what control was?

line 175: only one cytokine? and what is :”test compound”?

lines 177-178: “These data suggest that an anti- inflammatory effects of CLE at 100 and 1000 µg/mL might be due to cytotoxicity” – but on basis of which results such conclusion was drawn?

Figure legends should be formatted according to journal requirements.

Relative  mRNA expression should be replaced by relative gene expression, synthesis of mRNA is a result of gene transcription.

line 249: why investigations of level of proteins involved in cell cycle progression is called effects on immune response?

line 253: here and in whole text: “However, cyclin D2 and cyclin E2, cell cycle inducers, was increased…” – but what was increased? content? production? secretion?

  • Discussion:

lines 261-264: why sepsa is again evoked? How edible seaweed is connected with sepsa?

lines 272-277: the results of current study should be better connected with the given references that is it should be clearly stated that CLE subjected for the examination also contained such compounds;

  • Conclusions:

line 315: CLE rich in tannin and catechin…- unclear;

The conclusion concerning enhancing immune response is irrelevant to the examinations performed. It concerns only the regulation of cell cycle progression.

Author Response

Response to Referee’s Comments

We greatly appreciate the reviewer’s comment which have indeed helped to improve the quality of this manuscript. The detail is present as suggested in a newly revised manuscript and the changes have made as highlighted in “red font”. We hope our revision has improved the paper to a satisfactory level. This revised manuscript has undergone English language editing by MDPI.

Major comments:

  1. The inflammatory cytokines, enzymes, and mediators were evaluated. -the presence or production or both

We have changed the sentence from “The inflammatory cytokines, enzymes, and mediators were evaluated.” to “Expression and production of the inflammatory cytokines, enzymes, and mediators were evaluated.” (line 24-25)

  1. Results: CLE suppressed pro-inflammatory cytokines; IL-6 and TNF-α production and gene expression. – which gene expression?

We have changed the sentence from “CLE suppressed pro-inflammatory cytokines; IL-6 and TNF-α production and gene expression.” to “CLE suppressed expression and production of the pro-inflammatory cytokines IL-6 and TNF-α.” (line 25-26)

  1. What is connect between immune response and protein involved in regulation of cell cycle progression?

The immune response is the defense mechanism of the immune system against bacteria, viruses, and substances that appear foreign and harmful. Previous studies reported that cell cycle regulators (tumor protein p53 (p53), the cyclin-dependent kinase inhibitor p27 (p27), cyclin D2, and cyclin E2) are involved in the immune response. The p53 gene and protein are part of the innate immune system, and play a key role in defense against infectious diseases [5]. In addition, p27 played a role in immune-mediated inflammation in a glomerulonephritis model [6]. Moreover, cell cycle proteins (cyclin D and E) are involved in the homeostasis and integrity of the immune system [7]. (line 45-53)

4. The conclusion in the “Abstract” is not justified by results summarized in this section.

We agree with the reviewer that the conclusion in the “Abstract” is not justified by results summarized in this section. Thus, we have rewritten as suggested in “Abstract” section in line 29-31.

  1. Abstract improvement

We agree with the reviewer that the “Abstract” requires substantial improvement. Therefore, we have rewritten as suggested.

  1. Introduction improvement

We agree with the reviewer that Introduction” requires improvement. Therefore, we have rewritten as suggested in “Introduction” section in line 45-54 and added these 3 additional references in “References” section.

  1. Materials and methods improvement
  • Format

We have changed the format of “Materials and methods section as suggested.

  • ELISA is an acronym and should be written using Capital letters in the whole text

We have changed the word “Elisa” to “ELISA” throughout the manuscript as suggested.

  • Typo errors “0.4 ? M”

We have changed the word “0.4 ? M” to “0.4 M” as suggested.

  1. Results improvement
  • Format

We have changed the format of “Results section as suggested.

  • Concentration in mg/kg?

The content of phenolic compounds such as tannic acid,  rutin, isoquercetin, gallic acid and quercetin were present at mg of phenolic in 1 kg of CLE. However, we changed concentration in mg/g.  “The calculated amount of each major constituent found in CLE, as obtained from their respective calibration curves, are shown in milligrams of the phenolic compound per gram of CLE.” (line 73-74)

  • We agree with the reviewer and we have rewritten the sentence from “As shown in Figure 2, LPS at 1µg/mL significantly increased pro-inflammatory cytokine; IL-6, IL-1β, and TNFα compared control cells.” to “As shown in Figure 2, IL-6, IL-1β, and TNFα levels significantly increased in 1µg/mL LPS treated cells compared to that of control.” in line 81-82.

  • Please remove “;” moreover it is not clear what control was?

We agree with the reviewer and we have remove “;” symbol and rewritten the sentence from “As shown in Figure 2, LPS at 1µg/mL significantly increased pro-inflammatory cytokine; IL-6, IL-1β, and TNFα compared control cells.” to “As shown in Figure 2, IL-6, IL-1β, and TNFα levels significantly increased in cells treated with 1 µg/mL LPS compared with the control.” (line 85-87) In addition, the control cells in this study were incubated with free medium without LPS.

  • Only one cytokine? And what is: “test compound”?

We have changed the sentence from “Similarly, celecoxib (CX), nonsteroidal anti-inflammatory drug also decreased pro-inflammatory cytokine.” to “Similarly, celecoxib (CX), a non-steroidal anti-inflammatory drug, also decreased pro-inflammatory cytokines” (line 90-91)

Moreover, we have rewritten the sentence from “In addition, cell viability under test compound exposure was also examined.” to “In addition, cell viability in test compounds, including LPS, 0.1–1000 µg/mL, and CX was also examined.” (line 91-92)

  • Remove the sentence “These data suggest that an anti-inflammatory effects of CLE at 100 and 1000 µg/mL might be due to cytotoxicity.

We have changed this sentence as suggested from “These data suggest that an anti- inflammatory effects of CLE at 100 and 1000 µg/mL might be due to cytotoxicity. Thus, CLE at 10 µg/mL was selected because it significantly decreased inflammatory cytokines without cytotoxicity.” to “Due to the cytotoxicity of CLE at 1000 µg/mL, CLE at 10 µg/mL was selected because it significantly decreased inflammatory cytokines without cytotoxicity.” (line 94-95)

  • Figure legends format

We have changed the format of “Figure legends section as suggested.

  • Relative mRNA expression should be replaced by relative gene expression.

We have changed the word “mRNA” to “gene” throughout the manuscript as suggested.

  • Why investigations of level of proteins involved in cell cycle progression id called effects on immune response?

The immune response is the defense mechanism of the immune system against bacteria, viruses, and substances that appear foreign and harmful. Previous studies re-ported that cell cycle regulators (tumor protein p53 (p53), the cyclin-dependent kinase inhibitor p27 (p27), cyclin D2, and cyclin E2) are involved in the immune response. The p53 gene and protein are part of the innate immune system, and play a key role in de-fense against infectious diseases [5]. In addition, p27 played a role in immune-mediated inflammation in a glomerulonephritis model [6]. Moreover, cell cycle proteins (cyclin D and E) are involved in the homeostasis and integrity of the immune system [7]. (line 45-52)

References

[5] Levine AJ. P53 and The Immune Response: 40 Years of Exploration-A Plan for the Future. Int J Mol Sci. 2020;21(2):541.

[6] Ophascharoensuk V, Fero ML, Hughes J, Roberts JM, Shankland SJ. The cyclin-dependent kinase inhibitor p27Kip1 safeguards against inflammatory injury. Nat Med. 1998;4(5):575-80.

[7] Laphanuwat P, Jirawatnotai S. Immunomodulatory Roles of Cell Cycle Regulators. Front Cell Dev Biol. 2019;7:23.

  • However, cyclinD2 and cyclin E2, cell cycle inducers, was increased in CLE treated cells. – but what was increased? Content? Production? Secretion?

We have changed the sentence from “However, cyclin D2 and cyclin E2, cell cycle inducers, was increased in CLE treated cells.” to “However, the expression of cyclin D2 and cyclin E2, which are cell cycle inducer genes, was increased in CLE-treated cells.” (line 168-169)

  1. Discussion improvement
  • Why sepsa is again evoked? How edible seaweed is connected with sepsa?

Sepsis is an inflammatory disease mediated by the host immune response. Moreover, we have explained that tannic acid, major constituent in CLE, has an inhibitory effect on pro-inflammatory cytokines expression and production.

  • The results of current study should be better connected with the given references that is it should be clearly state that CLE subjected for the examination also contained such compound.

We have added the sentence “Thus, anti-inflammatory effect of CLE in this study might be due to tannic acid and catechin.” in line 193-194.

  1. Conclusion improvement
  • CLE rich tannin and catechin….-unclear

We have changed the sentence from “CLE rich tannin and catechin improves inflammatory status by inhibiting gene expression and production of pro-inflammatory cytokines and their mediators, and also suppressing apoptotic body formation and DNA damage.” to “CLE contains tannin and catechin, and it improves inflammatory status by inhibiting the gene expression and production of pro-inflammatory cytokines and their mediators, and also by suppressing apoptotic body formation and DNA damage.” (line 326-328)

  • Cell cycle regulators and immune response

As mention earlier that the cell cycle regulators play a key role in several biological processes, including DNA damage repair, cell death, cell differentiation, metabolism, and immune defense [7]. Thus, we have concluded that CLE enhances the immune response by modulating cell cycle regulators.

References

[7] Laphanuwat P, Jirawatnotai S. Immunomodulatory Roles of Cell Cycle Regulators. Front Cell Dev Biol. 2019;7:23.

Round 2

Reviewer 3 Report

The authors of the manuscript “The enhancing immune response and anti-inflammatory effects of Caulerpa lentillifera extract in RAW 264.7 cells” have addressed most of my concerns.

Minor issues:

Sentence should be rephrased for clarity.

Lane 15:  Background: Low grade Caulerpa lentillifera (CL) is the first by-product obtained from fresh-cut processing.

Comma is missing at the end of the sentence:

Lane 17:  CL was dried and extracted for determination of its active compounds and evaluation 16 of biological activities

Amounts instead of amount:

Lanes 71-72: The calculated amounts of each major constituent found in CLE, as obtained from their respective calibration curves, are shown in milligrams of the phenolic compound per gram of CLE.

Figure 2

The way the data is presented, it looks as if the following samples were analysed:

Sample 1: no treatment

Sample 2: only LPS

Sample 3: only CLE (0.1)

Sample 4: only CLE (1)

Sample 5: only CLE (10)

Sample 6: only CLE (100)

Sample 7: only CLE (1000)

Sample 8: only CX

This does not fit to the explanation in lanes 97-98: “RAW 264.7 cells were treated with different concentrations of CLE in the presence or absence of LPS for 24 h.”

If LPS is used in samples 3-8 as well, the sentence should be: “RAW 264.7 cells were treated with LPS in the absence or presence of different concentrations of CLE for 24 h.”

Which sample(s) show combined treatment with LPS and CLE?

Why is LPS sample missing in the bottom right panel of the Figure 2?

Statement “In addition, COX-2 enzymes also produce PGE2, a principal mediator of inflammation.” is incorrect, since PGH2 (product of COX-2 enzyme) is further processed by PGE Synthase to generate PGE2. The sentence should be corrected to reflect that.

Same comment refers also to the sentence “CLE also reduced PGE2 levels and its synthase enzyme, COX-2, similar to the action of NSAIDs.”

No need to explain abbreviation twice in the text – abbreviation should be explained only the first time it is used:

Lane 200: “…nitric oxide synthase (iNOS) and cyclooxygenase 2 (COX-2)…”

The authors KEPT (!?) the “marketing overstatement” sentence, which I already commented previously that it had to be REMOVED.

I will therefore repeat my comment from the first revision:

The effect is also measured in only one cell line making it unclear if it is cell line-specific or more general. Furthermore, higher CLE doses are highly cytotoxic and therefore the claim that “This magnificent effect of CLE makes it a new alternative approach as a natural anti-inflammation supplement.” is a clear overstatement and should be removed.

Author Response

Response to Referees Comments

We greatly appreciate the reviewer’s comment which have indeed helped to improve the quality of this manuscript. The detail is present as suggested in a newly revised manuscript and the changes have made as highlighted in “red font”. We hope our revision has improved the paper to a satisfactory level.

Reviewer #3

Minor comments:

  1. Sentence should be rephrased for clarity.

Lane 15:  Background: Low grade Caulerpa lentillifera (CL) is the first by-product obtained from fresh-cut processing.

  • As suggested, we have re-written the sentence as follow; Caulerpa lentillifera (CL) is a green seaweed, all the edible parts of CL were added value as functional ingredient.
  1. Comma is missing at the end of the sentence

Lane 17:  CL was dried and extracted for determination of its active compounds and evaluation 16 of biological activities

  • As suggested, we have added full stop symbol (.) at the end of a sentence.
  1. Amounts instead of amount
  • As suggested, we have replaced the word “amounts” instead of “amount” in line 72
  1. Figure 2
    • Edited figure
  • As suggested, we have edited the Figure 2 and re-written the sentence as follow; RAW 264.7 cells were treated with LPS in the absence or presence of different concentrations of CLE for 24 h.
    • Why is LPS sample missing in the bottom right panel of the Figure 2?
  • Figure 2D showed the cell viability in test compounds (CLE and CX). The objective of this figure was to confirm that the anti-inflammatory effects of test compounds in Figure 2A-C is not due to
  1. Statement “In addition, COX-2 enzymes also produce PGE2, a principal mediator of inflammation.” is incorrect
  • As suggested, we have re-written the sentence as follow; In addition, prostaglandin G2 is converted into prostaglandin H2 by COX enzymes and then activated prostaglandin synthases to generate PGE2, a principal mediator of inflammation. (line 193-195)
  1. Same comment refers also to the sentence “CLE also reduced PGE2 levels and its synthase enzyme, COX-2, similar to the action of NSAIDs.”
  • We have re-written the sentence as follow; CLE also reduced PGE2 levels and COX-2 enzyme. (line 198)
  1. No need to explain abbreviation twice in the text
  • We have checked and edited the abbreviation throughout the manuscript as suggested.
  1. Marketing overstatement
  • We have removed the sentence “This magnificent effect of CLE makes it a new alternative approach as a natural anti-inflammation supplement.” as reviewer’suggestion.

Reviewer 4 Report

Very Intereting study. Good work, congratulations

Author Response

Thank you for your consideration of this manuscript.

Respectfully yours,

Author team